# Aminopeptidase Activities Interact Asymmetrically between Brain, Plasma and Systolic Blood Pressure in Hypertensive Rats Unilaterally Depleted of Dopamine

**DOI:** 10.3390/biomedicines10102457

**Published:** 2022-10-01

**Authors:** Inmaculada Banegas, Isabel Prieto, Ana Belén Segarra, Francisco Vives, Magdalena Martínez-Cañamero, Raquel Durán, Juan de Dios Luna, Germán Domínguez-Vías, Manuel Ramírez-Sánchez

**Affiliations:** 1Department of Health Sciences, University of Jaén, 23071 Jaén, Spain; 2Department of Physiology, Faculty of Medicine, University of Granada, 18071 Granada, Spain; 3Department of Biostatistic, University of Granada, 18071 Granada, Spain; 4Department of Physiology, Faculty of Health Sciences, Ceuta, University of Granada, 18071 Granada, Spain

**Keywords:** dopamine, limbic system, medial prefrontal cortex, hippocampus, amygdala, brain asymmetry, hypertension, asymmetrical neurovisceral integration, proteinases, aminopeptidases, renin-angiotensin system

## Abstract

Brain dopamine, in relation to the limbic system, is involved in cognition and emotion. These functions are asymmetrically processed. Hypertension not only alters such functions but also their asymmetric brain pattern as well as their bilateral pattern of neurovisceral integration. The central and peripheral renin-angiotensin systems, particularly the aminopeptidases involved in its enzymatic cascade, play an important role in blood pressure control. In the present study, we report how these aminopeptidases from left and right cortico-limbic locations, plasma and systolic blood pressure interact among them in spontaneously hypertensive rats (SHR) unilaterally depleted of dopamine. The study comprises left and right sham and left and right lesioned (dopamine-depleted) rats as research groups. Results revealed important differences in the bilateral behavior comparing sham left versus sham right, lesioned left versus lesioned right, and sham versus lesioned animals. Results also suggest an important role for the asymmetrical functioning of the amygdala in cardiovascular control and an asymmetrical behavior in the interaction between the medial prefrontal cortex, hippocampus and amygdala with plasma, depending on the left or right depletion of dopamine. Compared with previous results of a similar study in Wistar-Kyoto (WKY) normotensive rats, the asymmetrical behaviors differ significantly between both WKY and SHR strains.

## 1. Introduction

Brain dopamine maintains a close relationship with areas involved in cognitive functions and emotional control such as the medial prefrontal cortex, hippocampus and amygdala, included in the limbic system [1,2,3,4,5]. Such functions are processed asymmetrically [6]. In fact, the coexistence between dopamine, Parkinson’s disease, altered emotional state and cerebral asymmetry is well known [7,8,9]. Hypertension not only alters those cognitive and emotional processes but can also modify the bilateral asymmetric model, not only of brain function per se but also of the asymmetric model of neurovisceral integration [10,11,12,13]. Among others, the renin-angiotensin-aldosterone system (RAAS) plays an essential role in controlling blood pressure [14]; in fact, common antihypertensive treatments include inhibitors of the system at various levels [15] which have been also suggested for the prevention and treatment of neurodegenerative diseases such as Alzheimer’s disease [16,17] or Parkinson’s disease [18]. Within the RAAS, proteolytic enzymes and, in particular, some aminopeptidases, play a relevant role [19]. With this research, we analyze how these aminopeptidases, measured from left and right cortico-limbic locations (medial prefrontal cortex, mPFC, hippocampus, HC, and amygdala, AM), plasma and systolic blood pressure interact among them in spontaneously hypertensive rats (SHR) unilaterally depleted of dopamine. In addition to blood pressure research, SHR is a frequent animal model used to study brain disorders such as attention deficit/hyperactivity disorder [20], Schizophrenia [21] and vascular brain disorders among many others [22]. In this context, other contributions are interesting, such as the relationship between dopamine and cognitive function [23], mitochondrial dysfunction in schizophrenia [24], or schizophrenia in the dissociative model [25].

Brain functions and bidirectional neuro-visceral integrative processes are virtually asymmetric [26] but this bilateral pattern is dynamically modified depending on physiological and pathological conditions [26], as well as depending on changing environmental factors, even going so far as to change the side of predominance [27]. However, despite extensive research on brain asymmetry, its functional significance remains largely unknown [26,28,29,30].

Neuropeptides and peripheral peptides take an important role in many of those processes, being the functional status of these peptides is regulated in part by proteinases [27,31]. In the brain, this regulation is mainly carried out by aminopeptidases as its most abundant proteinases [32]. This is the rationale we used to analyze the bilateral distribution of leucine-aminopeptidase in the brain, a study that led to the first description of a brain asymmetry for proteinase activity, particularly in the frontal cortex of rats [33]. Simultaneously, the activities of two classical metabolic enzymes, lactate dehydrogenase and glutamate-oxalacetate aminotransferase, were measured in the same cerebral region and no significant difference was recorded in these activities between both sides of the rat brain which may suggest a specific functional role for that proteolytic activity [34].

A clear illustrative example of the functional role exerted by proteinases can be observed in the role they play in the enzymatic cascade of the RAAS [35] (Figure 1). The proteolytic activity of an enzyme results in the inactivation of its substrate and simultaneously in the activation of the new peptide resulting from such hydrolysis, with different functions from those from which it came. Therefore, its analysis offers us a dynamic view of its function under certain experimental conditions.

Despite the initial conviction of “a neuron a neurotransmitter”, nowadays the coexistence of different types of neurotransmitters in the same neuron has become evident [36]. For example, it has been demonstrated the coexistence in dopaminergic neurons of neuropeptides such as cholecystokinin and neurotensin [37], which can also interact with other neuropeptides such as angiotensins or enkephalins, all of them related to emotional processing and cardiovascular regulation [38,39]. The modification of any of these peptides may affect the function of all the others, including the bilateral interactions established between them. Dopamine plays a relevant role in emotional processing in which the cortico-limbic areas of the brain are involved. Brain dopaminergic asymmetry in physiological and pathological conditions such as Parkinson’s disease is also well known. Its modification will affect the functions performed by the various neurotransmitters that interact with it [39]. 

There is an increase in the levels of anxiety and depression in hypertensive patients [40] and in genetically hypertensive animals [41], emotional alterations in which disruptions in neuro-visceral communication may also be involved [42]. The coexistence between Parkinson’s disease and hypertension has also been discussed [43] and a connection between depression, neurochemical asymmetry and cardiovascular function has been postulated [44]. 

Recently, we have described an asymmetric interaction of aminopeptidase activities between cortico-limbic areas, such as the prefrontal cortex, amygdala and hippocampus, of the left and right hemisphere, between these left or right brain areas and the same plasmatic activities and finally, between the left or right brain areas and systolic blood pressure (SBP) levels, following unilateral left or right dopamine depletions of the nigrostriatal system, in normotensive WKY rats as well as in their corresponding left or right sham controls [39]. Therefore, since emotional processing differs in hypertension and the bilateral neuro-visceral connection may be modified, it is necessary to analyze what happens in hypertensive conditions. In the present study, we report how the same brain locations, plasma and SBP interact in spontaneously hypertensive rats (SHR) unilaterally depleted of dopamine. The study comprises left and right sham and left and right dopamine-depleted (lesioned) rats as research groups. A global comparative analysis between both WKY and SHR strains of rats is also performed.

## 2. Materials and Methods

### 2.1. Experimental Groups

For the present research, a total of 40 male hypertensive rats (spontaneously hypertensive rats (SHR) from Charles River Laboratories, Barcelona, Spain). They were randomly separated into 4 groups of 10 animals each. Depending on the left or right brain hemisphere that we selected, the groups were called: sham left (SL), sham right (SR), lesioned left (LL) and lesioned right (LR). The left or right sham animals were obtained by the stereotaxic injection of saline into the left or right striatum. The animals in which the left or right depletion of dopamine was carried out (lesioned) were achieved by the stereotaxic injection of the neurotoxin 6-hydroxidopamine (6-OHDA) into the left or right striatum [39].

The animals correctly left or right depleted of dopamine demonstrated, four weeks after 6-OHDA injections, an outstanding turning behavior towards the hemisphere having less dopamine content [45]. When the animals recovered from surgery, systolic blood pressure (SBP) was measured by the plethysmographic method as previously reported in detail [46] (Figure 2). The rats were then sacrificed on the same day and blood samples as well as samples from the left or right medial prefrontal cortex (mPFC), left or right hippocampus (HC) and left or right amygdala (AM) were obtained from each group and frozen until assayed. All the experiments and the care of animals were performed in agreement with the European Communities Council Directive 86/609/EEC. Approved by the bioethic committee of the University of Jaén on 20 November 2006 with number 140.

### 2.2. 6-Hydroxydopamine and Saline Intrastriatal Injections

The animals, previously anesthetized with equithensin (42.5 g/L chloral hydrate dissolved in 19.76 mL ethanol, 9.72 g/L Nembutal^®^, 0.396 g/L propylene glycol and 21.3 g/L magnesium sulfate in distilled water) (2 mL/kg body weight), were fixed to a stereotaxic (David Kopf Instruments, Palo Alto, CA, USA). In agreement with the Paxinos and Watson atlas [47], the coordinates used to inject saline or the neurotoxin 6-OHDA into the striatum were: AP 0 mm, L or R 3 mm and H-5 mm. For left or right lesioned rats, 4 μL of 6-OHDA (8 mg dissolved in 1 mL of cold saline with 0.02% ascorbic acid to inhibit oxidation) was injected. For left or right sham rats, 4 μL of saline with 0.02% ascorbic acid was injected.

### 2.3. Obtaining Brain and Plasma Samples

Four weeks after the injections of saline or 6-OHDA, to confirm that the lesions had been carried out correctly, the turning behavior of the animals was measured [46]. Then, under anesthesia, blood samples were obtained from the left ventricle. Plasma, obtained by centrifugation (10 min at 3000× *g*), was used for enzyme and protein determinations in triplicate. After obtaining the blood samples, the animals were perfused with saline (also through the left ventricle) and their brains were obtained in less than 1 min and cooled in dry ice. Subsequently, according to the Paxinos and Watson atlas [47], left and right samples of mPFC, HC and AM were obtained. The coordinates for mPFC were between 12.70 mm and 11.20 anterior to the interaural line (AIL), between 7.12 and 5.40 mm AIL for HC, and between 7.12 and 5.40 mm AIL for the AM [39].

### 2.4. Aminopeptidase Activities and Protein Determinations

The method for the measurement of aminopeptidases and proteins has already been described [31]. In summary, once the tissue samples were defrosted, they were homogenized (in 400 μL of 10 mM HCl-Tris buffer, pH 7.4) and centrifuged (100,000× *g* for 30 min at 4 °C). To separate membrane proteins, the pellets were again homogenized in HCl-Tris buffer (pH 7.4) plus 1% Triton-X-100 and centrifuged (100,000× *g*, 30 min, 4 °C). In order to remove the detergent Triton-X-100, the obtained supernatants were shaken in an orbital rotor (2 h at 4 °C) together with the polymeric adsorbent Bio-Beads SM-2 (100 mg/mL). These supernatants were then used to determine, in triplicate, membrane-bound aminopeptidase activities and proteins. Aminopeptidases were measured by fluorimetry [31]. In summary, AlaAP and CysAP were measured using the aminoacil-β-naphtylamides (aaNNap) AlaNNap and CysNNap as substrates: 10 μL of each supernatant were incubated 30 min at 25 °C with 1 mL of the substrate solution: 2.14 mg/100 mL AlaNNap or 5.63 mg/100 mL CysNNap, 10 mg/100 mL bovine serum albumin (BSA), and 10 mg/100 mL dithiothreitol (DTT) in 50 mM of phosphate buffer, pH 7.4, for AlaAP and 50 mM HCl–Tris buffer, pH 6, for CysAP. AspAP was measured using AspNNap as substrate: 10μL of each supernatant was incubated for 120 min at 37 °C with 1 mL of the substrate solution (2.58 mg/100 mL AspNNap, 10 mg/100 mL BSA and 39.4 mg/100 mL MnCl_2_ in 50 mmol/L HCl–Tris buffer, pH 7.4). GluAP was determined using GluNNap as substrate: 10 μL of each supernatant was incubated during 120 min at 37 °C with 1 mL of the substrate solution (2.72 mg/100 mL GluNNap, 10 mg/100 mL BSA, 10 mg/100 mL DTT and 0.555 g/100 mL CaCl_2_ in 50 mmol/L HCl–Tris, pH 7.4). In order for the enzymatic reactions to stop, 1 mL of 0.1 mol/L of acetate buffer at pH 4.2 was added. The β-naphthylamine, released by enzymatic activity, was measured, by fluorimetry, at 412 nm emission wavelengths with an excitation wavelength of 345 nm. The method allows determinations in the picomolar range. Proteins were measured by the method of Bradford [48], using BSA as a standard. Specific membrane-bound brain aminopeptidase activities were expressed as nmol of AlaNNap, CysNNap, AspNNap, or GluNNap hydrolyzed per min per mg of protein. For determinations of plasma aminopeptidase activities, the procedure was modified as follows: 25 μL of plasma was incubated for 120 min at 37 °C with 1 mL of the substrate solutions previously indicated. Plasma aminopeptidase activities were expressed as pmol of the corresponding aaNNap hydrolyzed per min per mg of protein [46]. Fluorogenic assays were linear with respect to the time of hydrolysis and protein content.

### 2.5. Statistical Analysis

The differences between hemispheres and groups were calculated using a two-way analysis of variance. Post hoc comparisons were performed with LSD tests. *p*-values below 0.05 were considered significant. To study the intra- and inter-hemispheric levels of correlation between neuropeptidase activities of the mPFC, HC and AM, between them and plasma aminopeptidases and between them and SBP, Pearson’s coefficient of correlation was computed. Computations were performed using SPSS 13.0 and STATA 9.0 (STATA Corp., College Station, TX, USA). *p*-values below 0.05 were considered significant.

## 3. Results

### 3.1. Systolic Blood Pressure in Hypertensive Rats

While SBP levels did not vary in the left lesioned (left dopamine-depleted) animals (LL) in comparison to their sham left (SL) controls, animals with the DA-depletion of the right hemisphere (LR) significantly increased (*p* < 0.001; ***) SBP levels compared to their corresponding sham controls (SR). The right SR and LR groups showed significantly (*p* < 0.001; +++) lower SBP levels than the left SL or LL groups, respectively (Figure 2). 

### 3.2. Aminopeptidase Activities in Brain and Plasma

In general, the data show a great variability depending on the enzymatic activity analyzed and the brain or plasmatic location (Figure 3). In plasma, a significant reduction is observed for AlaAP (*p* < 0.001), CysAP (*p* < 0.001) and AspAP (*p* < 0.01) or a tendency to decrease for GluAP in animals with left (LL) depletion of DA in comparison with their SL controls. In sharp contrast, there is a significant increase (*p* < 0.001) for GluAP, AlaAP, and AspAP or a tendency to increase for CysAP in right DA-depleted (LR) animals compared to their SR controls. At the cerebral level, the results are variable depending on the location, but in general, in most groups, less aminopeptidase activity is observed on the right side than on the left. 

### 3.3. Sham Left Correlations in Spontaneously Hypertensive Rats

The significant correlations in the group of sham left animals are indicated in Table 1. Except for a negative correlation between the left prefrontal cortex and the left hippocampus, the rest of the intra-left hemisphere correlations were abundant and positive with a high level of significance. In contrast, the number of intra-right hemisphere correlations was much lower, although the positive ones also predominated. The inter-hemispheric correlations are divided between five negative and three positive with a moderate level of significance, involving the three brain locations analyzed. The right hemisphere showed only negative correlations with plasma aminopeptidases. The left hemisphere showed two positive and one negative correlation with plasma. No intraplasmatic correlations or correlations between brain aminopeptidases and SBP levels were observed. 

### 3.4. Sham Right Correlations in Spontaneously Hypertensive Rats 

The significant correlations in the group of sham right animals are indicated in Table 2. In contrast to the SL group, the sham right animals present a greater number of right intra-hemispheric correlations, with a higher level of significance than the left intra-hemispheric ones, but mainly, in this group, there is a marked increase in inter-hemispheric correlations largely positive and highly significant. Also noteworthy is the greater number of correlations between the right hemisphere and plasma, compared to the low number of correlations between the left hemisphere and plasma. As was the case in SL animals, there are no correlations between brain and SBP levels and there is only a positive intra-plasma correlation. 

### 3.5. Lesioned Left Correlations in Spontaneously Hypertensive Rats

The significant correlations in the group of animals with left DA depletion (LL) are indicated in Table 3. In this group, a high number of positive intra-hemispheric correlations in the left hippocampus are observed, with a high level of significance. Three negative correlations between the left amygdala and the left hippocampus are also observed. The right intra-hemispheric correlations are also mainly positive and a negative correlation is observed, also between the amygdala and the hippocampus. In the inter-hemispheric correlations, similar to what occurred in the SL animals, there is a remarkable number of negative correlations, comparable to the positive ones and involving the three areas analyzed. The number of correlations, all of them positive, is similar between the left hemisphere and plasma than between the right hemisphere and plasma, but in the latter, the level of significance is much higher. Unlike the two previous groups, there is a high number of highly significant intra-plasma correlations. Finally, it is noteworthy that this group of LL is the only one in which correlations between the brain and SBP levels appear, three being negative with the left hemisphere and one positive with the right hemisphere. 

### 3.6. Lesioned Right Correlations in Spontaneously Hypertensive Rats

The significant correlations in the group of animals with right DA depletion (LR) are indicated in Table 4. A clear predominance is observed in the intra-left hemisphere correlations, most of them being positive with a high level of statistical significance. There is a smaller number of intra-right hemisphere correlations in which two negative correlations are also observed. Compared to the group of left-lesioned animals, the right ones present less inter-hemispheric correlations, with a similar number of negative correlations. The number of intra-plasma correlations is similar to that of the LL animals, all of them positive but with a lower level of significance. While the left prefrontal cortex exhibits one positive correlation with plasma, the right hippocampus exhibits three negative correlations with plasma. In this group, no correlations with SBP are observed.

### 3.7. Comparisons between Groups of Spontaneously Hypertensive Rats

In Figure 4 we observe that the comparison of the total number of significant correlations between SL vs. LL (Figure 4A) is opposite to the comparison between SR vs. LR (Figure 4B). In the comparison between LL vs. LR (Figure 4C), a greater number of Intra-Left than Intra-Right correlations is observed in both LL and LR, and a greater number of inter-hemispheric correlations in LL than in LR (Figure 4C). In the correlations between plasma and brain in the left groups SL and LL (Figure 4D) there are more Plasma-Left and Plasma Right correlations in LL than in SL (Figure 4D). This pattern is opposite to that observed in the right group’s SR and LR (Figure 4E). When comparing LL vs. LR (Figure 4F) there is a higher number of Intra-Left and Intra-Right correlations in LL than in LR (Figure 4F). With systolic blood pressure (SBP), significant correlations are only observed in the LL group (Figure 4G–I). 

Except for the intra-location correlations (intra-P, intra-H, intra-A or intra-Pl), Figure 5 represents all the significant correlations that were obtained in all the groups analyzed corresponding to Table 1,Table 2,Table 3,Table 4. The pattern of correlations differs between all the groups studied. In the sham left group, we can highlight the positive correlations between left mPFC and left hippocampus (A) and negative correlations between left hippocampus and right amygdala (B). In the sham right group, we can highlight the global predominance of positive correlations, particularly between the left amygdala and left hippocampus and between right mPFC and right amygdala (A), but especially between the left and right amygdala and between left hippocampus and right amygdala (B). In animals with a left lesion, we highlight the negative correlations between the left amygdala and left hippocampus, the positive correlations between plasma and left amygdala or mPFC, and positive correlations between plasma and right mPFC (A). In this group, we also highlight the positive correlation between the left amygdala and right hippocampus, both positive and negative between the left amygdala and right amygdala and negative between left mPFC and right amygdala (B). However, especially noteworthy is that in hypertensive animals, this is the only group that correlates the brain with SBP, specifically the negative correlation with the left amygdala and positive with the right (B). In the group of animals with a right lesion, we can highlight the positive correlations between the left amygdala and left hippocampus, positive between left mPFC and plasma, negative between right hippocampus and plasma (A), and negative between left hippocampus and right amygdala (B). 

## 4. Discussion

Regarding the role in the emotional processing of brain asymmetry and neurovisceral asymmetry, considering the role of the interaction of DA with peptidergic or not peptidergic neurotransmitters, and with respect to the role of blood pressure levels, we can highlight some previous contributions: It has been described that there is an increase in anxiety and depression in hypertensive humans and SHR [40,41]. Emotional processing differs between normotensive and hypertensive patients [10]. The mPFC is important in such emotional processing [51]. Cognitive alterations have been described in hypertensive conditions in both animals and humans [52,53]. It is also necessary to take into account the existence of different functional consequences after left or right hemispheric damage, such as lower sensorimotor function after left damage and better cognitive function after left than after right damage [54]. On the other hand, cognitive alterations have been described in Parkinson’s disease that involves the hippocampus [55] and it has also been described that the amygdala participates in cardiovascular regulation [56]. Emotional alterations have also been described in hypertensive patients, alterations in which disruptions in neuro-visceral communication may be involved [42]. The consequences of the coexistence of Parkinson’s and hypertension have also been described, raising the possibility of the possible benefit of antihypertensive treatments in Parkinson’s patients [43]. Following 6-OHDA administration, corticotropin-releasing hormone and neurotensin mRNA expression is decreased in the amygdala and the bed nucleus of stria terminals [57]. Dopamine depletion causes asymmetric responses depending on gender and type of behavior [58]. Postmortem human brains demonstrate an asymmetry in opiate expression in the anterior cingulate cortex, an area involved in emotional and pain processing [59]. After DA depletion with 6-OHDA, CCK levels decrease more in SHR than in WKY [60]. A lower amount of DA in Am, Hc and mPFC of SHR than WKY has also been described [61]. As previously indicated, DA coexists with CCK and NT [37]. In addition, more DA receptors have also been described in SHR than in WKY [62] and brain angiotensin II, by means of its AT1 receptors, regulates dopaminergic neurotransmission [63], clearly involving the RAAS in neurodegenerative diseases [64].

After analyzing the present results, it is clear that the behavior of the SBP clearly differs between normotensive WKY [39] and SHR animals. While in WKY, the SBP does not vary after the left or right lesion, in the SHR the pressure does not vary after the left lesion but increases after the right one. In addition, the SR and LR groups have lower SBP than the SL and LL groups (Figure 2). 

Regarding the role of angiotensin peptides: Ang II is the main hypertensive peptide; Ang III has similar or less hypertensive properties than Ang II and is the main brain Ang peptide that stimulates the secretion of vasopressin; Ang IV, without hypertensive functions, is involved in the regulation of brain local blood flows and has an anxiolytic effect acting on the amygdala; Ang 2–10 may counterbalance the hypertensive effect of Ang II and Ang III in the brain [35]. According to the results obtained in plasma (Figure 3) and the role of aminopeptidases in RAAS (Figure 1), in LL there is a reduction of GluAP and AlaAP which could suggest a lower metabolism of Ang II and Ang III increasing consequently their hypertensive action. In LL there is a reduction of CysAP, increasing, therefore, the hypertensive action of vasopressin and also diminishing AspAP activity which could reduce the formation of Ang 2–10 moderating its hypertensive counterbalancing. In contrast, in LR there is a marked increase in GluAP and AlaAP which suggests the higher formation of Ang III and Ang IV with which we can think of a lower hypertensive effect than in the LL group. In LR there is no change in vasopressinase (CysAP) activity and an increase in AspAP raising the formation of Ang 2–10 with a possible counterbalance of Ang II and Ang III. Taken together, our results in plasma would be compatible with higher SBP in LL and lower SBP in LR.

In agreement with Zhai and Feng [60], the present results show that just the surgical injury produced by the introduction of the cannula and the subsequent injection of saline differs depending on whether we carried out the surgery in the left or right hemisphere. In this sense, in the present study with SHR, it is remarkable the large number of inter-hemispheric correlations involving the amygdala in the sham right animals compared to the sham left (Figure 5). This pattern clearly differs when we selectively deplete DA in the left or right hemisphere, showing that this new factor modifies the entire left or right pattern of correlations that we observed in the left or right sham animals.

Especially interesting is the behavior of the correlations between left or right hemispheric locations, SBP and plasma (Figure 6 and Figure 7), compared with the results previously obtained with normotensive WKY animals [39]. The SBP mainly correlates with the LL group in WKY and exclusively with the LL group in SHR. In WKY LL, the right amygdala correlates positively with SBP but the left hippocampus correlates negatively. In SHR LL, the pattern is similar: the right amygdala correlates positively with SBP but the left amygdala correlates negatively. These results highlight the importance of left DA depletion as well as the relevant role of the amygdala in cardiovascular regulation [56] in which there could be a balance or imbalance between the regulatory role of the left and right amygdala in the control of blood pressure. This suggestion could be supported if we analyze individually the consequences of each significant correlation involving the amygdala in LL SHR (Figure 6). In these LL animals, there is a positive correlation between GluAP of the right amygdala and the levels of SBP: the higher GluAP, the higher SBP. GluAP metabolizes Ang II to Ang III (Figure 1) which may imply an increase in SBP. There is a negative correlation between AspAP of the left amygdala and SBP, the higher AspAP, the lower SBP. AspAP metabolizes Ang I to Ang 2–10 which may counterbalance the effects of Ang II and Ang III contributing to a reduction of SBP. There is a negative correlation between CysAP of the left amygdala and SBP, the higher CysAP, the lower SBP, this is in agreement with a reduction of vasopressin due to the high levels of CysAP also contributing to lower SBP. There is also a slight significant correlation between GluAP of the left amygdala and SBP, the higher GluAP, the lower SBP. In this case, the increase in Ang III in the left amygdala, in contraposition with the positive correlation of GluAP of the right amygdala with SBP, would imply lower SBP. Taking together, in LL SHR, the right amygdala would be related to an increase in SBP and the left would be a decrease.

In WKY [39], the amygdala also has a relevant role correlating significantly with SBP. In SL, CysAP of the left amygdala presents a negative correlation with SBP, the higher CysAP, the lower SBP. In SR, GluAP of the right amygdala correlates positively with SBP, the higher GluAP, the higher SBP. In LL, GluAP of the right amygdala correlates negatively with SBP, the higher GluAP, the lower SBP and also in this group, CysAP of the right amygdala has a highly significant positive correlation with SBP, the higher CysAP, the higher SBP. In conclusion, these results support the role of the amygdala in cardiovascular control and extend it to the concept of bilateral asymmetrical control. 

With plasma, the pattern is reversed in LR SHR animals (Figure 7): While the left mPFC correlates positively with plasma GluAP, the right hippocampus correlates negatively with plasma aminopeptidases. In WKY, the negative correlation between the right hemisphere and plasma in the LR is maintained, but there is no correlation with the left hemisphere in this group. In the WKY LL, there is no specific pattern. We can therefore affirm that the response to left or right DA depletion is different in normotensive WKY rats than in SHR (Figure 8).

In general terms, there is a greater number of correlations in SHR than in WKY. In comparison with WKY, in SL SHR the left intra-hemispheric correlations increase, and the inter-hemispheric ones are inverted to negative, however, in SR SHR, inter-hemispheric and right intra-hemispheric increase. As occurred in the SL, in the SHR LL, compared to the WKY LL, the left intra-hemispheric correlations increase and the inter-hemispheric ones are inverted, but with greater intensity. In the LR, the left intra-hemispheric correlations increase in SHR, and the inter-hemispheric correlations are inverted with respect to the WKY. If we look at Figure 8, it seems clear that left DA depletion (LL) is the one that induces a greater number of correlations, particularly in SHR. It is also necessary to highlight that in WKY, all groups correlate between brain and SBP, especially after left DA depletion. In SHR, only after left DA depletion do brain-SBP correlations appear.

In conclusion, the present results with SHR and the previous ones with WKY demonstrate that selective unilateral depletion of a specific neurotransmitter changes radically the bilateral pattern of response, in these cases involving proteinase activities, suggesting functional changes depending on the bilateral consequences. These results reveal a high complexity in the bilateral response to physiological and pathological changes in the organism, without being able to establish yet a specific pattern of response. Perhaps, the most interesting results suggest an important role for the asymmetrical functioning of the amygdala in cardiovascular control, particularly when the left hemisphere was depleted of dopamine and an asymmetrical behavior in the interaction between the medial prefrontal cortex, hippocampus and amygdala with plasma, depending on the left or right depletion of dopamine which supports an asymmetric neurovisceral integration of the organism. In general terms, our results lead us to speculate on the concept of cerebral asymmetry, or rather neuro-visceral asymmetry, as an intrinsic property of the organism whose intensity or modification of the predominant side depends on multiple environmental, physiological and/or pathological factors [26,27]. This gives rise to a variable bilateral dynamic process in which many questions remain to be answered, such as what is the functional advantage of asymmetry over symmetry?; what emotional consequences derive from the bilateral cerebral and neuro-visceral biochemical modifications observed?; or how to connect the consequences of these modifications with the blood pressure control? However, it is necessary to indicate that the present study lacks parallel investigations of emotional behavior that could be interesting to understand the functional significance of brain asymmetry involving aminopeptidases. In addition, unilateral lesions in the analyzed areas could provide interesting suggestions about the role of these areas and aminopeptidases in blood pressure control. It would also be interesting to look into the analysis of the bilateral behavior of interactions between the limbic zones after treatments with RAAS inhibitors.

## Figures and Tables

**Figure 1 biomedicines-10-02457-f001:**
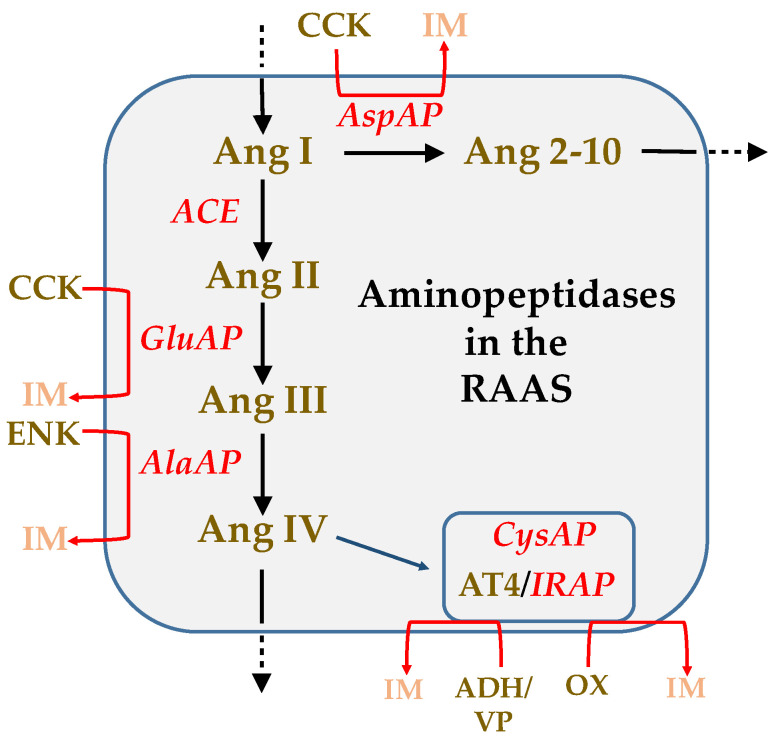
Schematic representation of a part of the renin-angiotensin-aldosterone system (RAAS) in which the aminopeptidase activities determined in the present work are involved. Angiotensin I (Ang I), produced by the action of renin on angiotensinogen, is hydrolyzed by aspartate aminopeptidase (AspAP) to produce Ang 2–10. AspAP can also metabolize cholecystokinin (CCK) to an inactive metabolite (IM). Moreover, Ang I, due to the action of the angiotensin-converting enzyme (ACE), produces Ang II, which is hydrolyzed to Ang III through the action of glutamate aminopeptidase (GluAP), which can also act on CCK to produce an IM. Ang III, through alanine aminopeptidase (AlaAP), will give rise to Ang IV but can also inactivate enkephalin (ENK). Ang IV binds to the AT_4_ receptor that has been identified as the insulin-regulated aminopeptidase (IRAP), also known as CysAP, oxytocinase (which inactivates oxytocin, OX), or vasopressinase (which inactivates vasopressin, VP or antidiuretic hormone, ADH) [35].

**Figure 2 biomedicines-10-02457-f002:**
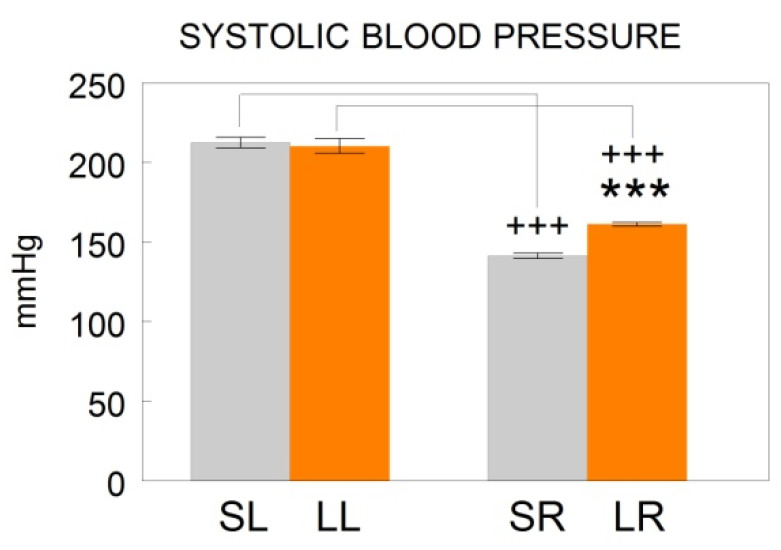
Mean ± SEM levels of systolic blood pressure values expressed in mmHg after saline injection into the left (sham left; SL; *n* = 10) or right (sham right; SR; *n* = 10) striatum or after 6-hydroxydopamine injection into the left (lesioned left; LL; *n* = 10) or right (lesioned right; LR; *n* = 10) striatum of spontaneously hypertensive adult male rats. *** indicates a significant difference of *p* < 0.001 between sham versus lesioned animals in the same side. +++ indicates a significant difference of *p* < 0.001 between sham left versus sham right or between lesioned left versus lesioned right animals. Modified from [46].

**Figure 3 biomedicines-10-02457-f003:**
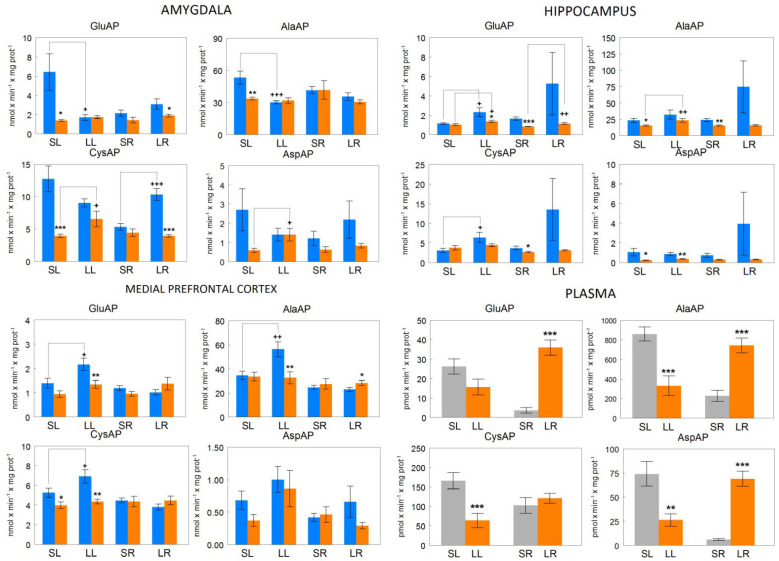
Glutamyl- (GluAP), alanyl- (AlaAP), cystinyl- (CysAP) and aspartyl- (AspAP) aminopeptidase activities, in their membrane-bound forms, expressed as nmol of L-Glu-β-naphthylamide, L-Ala-β-naphthylamide, L-Cys-β-naphthylamide and L-Asp-β-naphthylamide hydrolyzed per min per mg of proteins in the left (blue bars) or right (rose bars) side of the amygdala, hippocampus, medial prefrontal cortex and as pmol of the same substrates hydrolyzed per min, per mg of proteins in plasma of sham left (SL) or sham right (SR) and lesioned left (LL) or lesioned right (LR) adult male spontaneously hypertensive rats. In plasma, gray bars represent sham groups (SL, SR) and rose bars represent lesioned groups (LL, LR). Each group represents the mean ± SEM values for 10 animals. Asterisks (*) indicate the level of statistical significance between the left and right side of the brain or between sham vs. lesioned animals in plasma. Crosses (+) indicate statistical difference on the same side, between sham vs. lesioned rats. Single (*/+) (*p* < 0.05), double (**/++) (*p* < 0.01) or triple symbol (***/+++) (*p* < 0.001). Preliminary partial data of AlaAP from mPFC [49], CysAP from mPFC [50] and GluAP from plasma [46] and mPFC [45] were previously reported.

**Figure 4 biomedicines-10-02457-f004:**
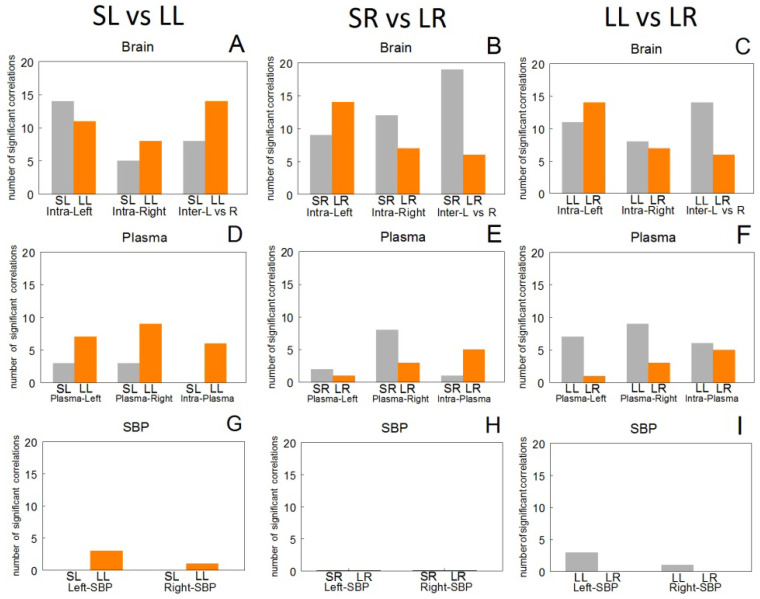
Total number of intra- (Intra-Left or Intra-Right) and inter-hemispheric (Inter-Left vs. Right) significant correlations between brain aminopeptidase activities (Brain) (**A**–**C**); between plasma aminopeptidase activities versus left (Plasma-Left) or right (Plasma-Right) brain aminopeptidase activities and Intra-Plasma aminopeptidase activities between themselves (Plasma) (**D**–**F**); between left (Left-SBP) or right (Right-SBP) aminopeptidase activities versus systolic blood pressure (SBP) in Sham Left (SL), Sham Right (SR), Lesioned Left (LL) or Lesioned Right (LR) spontaneously hypertensive rats (**G**–**I**).

**Figure 5 biomedicines-10-02457-f005:**
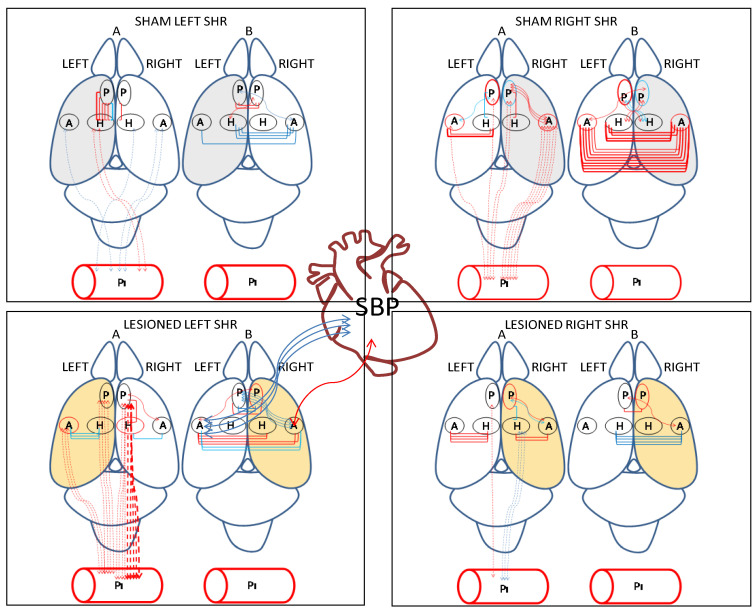
Schematic representation of the observed significant correlations: between brain aminopeptidase activities; between plasma aminopeptidase activities versus brain aminopeptidase activities; between brain aminopeptidase activities versus systolic blood pressure (SBP) in the four groups studied of spontaneously hypertensive rats. Positive correlations in red and negative correlations in blue. Continue lines: significant correlations between brain locations and between brain versus SBP. Dotted lines: significant correlations between brain versus plasma. The thickness of lines is proportional to the degree of significance. Above rat brain: A, intra-hemispheric and brain vs. plasma correlations, B, inter-hemispheric correlations. The hemisphere was injected with saline for sham animals in grey. The hemisphere was injected with 6-OHDA for lesioned animals in light orange. P, medial prefrontal cortex; A, amygdala; H, hippocampus; Pl, plasma.

**Figure 6 biomedicines-10-02457-f006:**
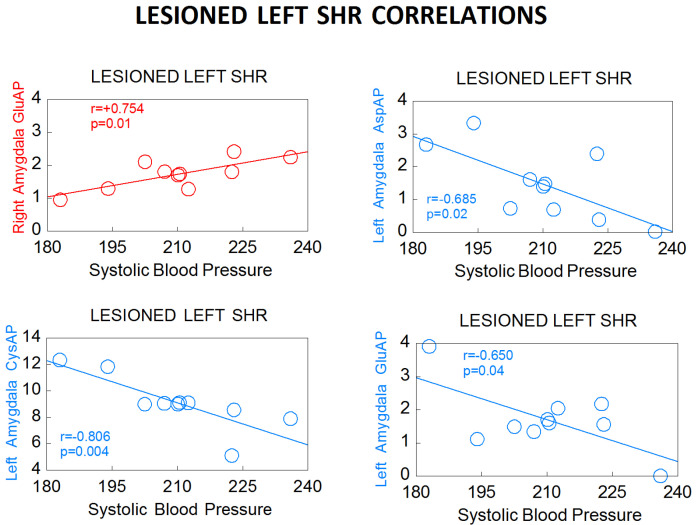
Significant correlations between aminopeptidase activities expressed as nmol of the corresponding L-amino acid-β-naphthylamide, hydrolyzed per min per mg of proteins in the left (blue) or right (red) amygdala and systolic blood pressure in lesioned left spontaneously hypertensive rats (SHR) (see Table 3). Negative correlations are in blue. Positive correlations are in red. Pearson’s correlation coefficients (r) and p values are indicated in each figure.

**Figure 7 biomedicines-10-02457-f007:**
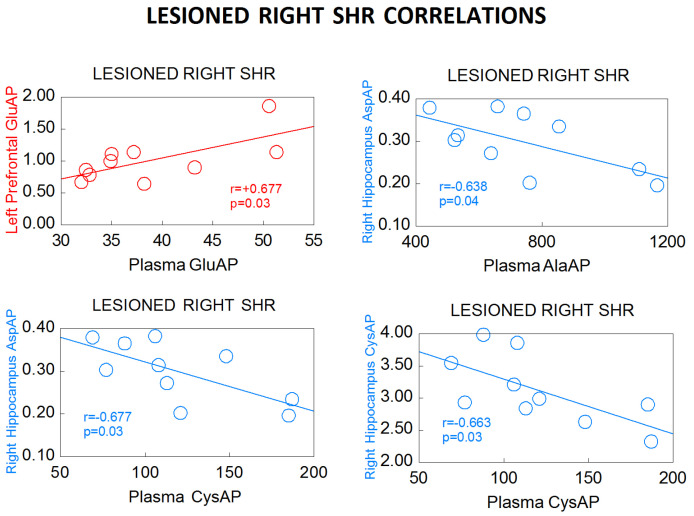
Significant correlations between aminopeptidase activities expressed as nmol of the corresponding L-amino acid-β-naphthylamide, hydrolyzed per min per mg of proteins in the left prefrontal cortex (red) or right hippocampus (blue), and plasma aminopeptidase activities, expressed as pmol of the corresponding L-amino acid-β-naphthylamide, hydrolyzed per min per mg of proteins, in lesioned right spontaneously hypertensive rats (SHR) (see Table 4). Negative correlations are in blue. Positive correlations are in red. Pearson’s correlation coefficients (r) and p values are indicated in each figure.

**Figure 8 biomedicines-10-02457-f008:**
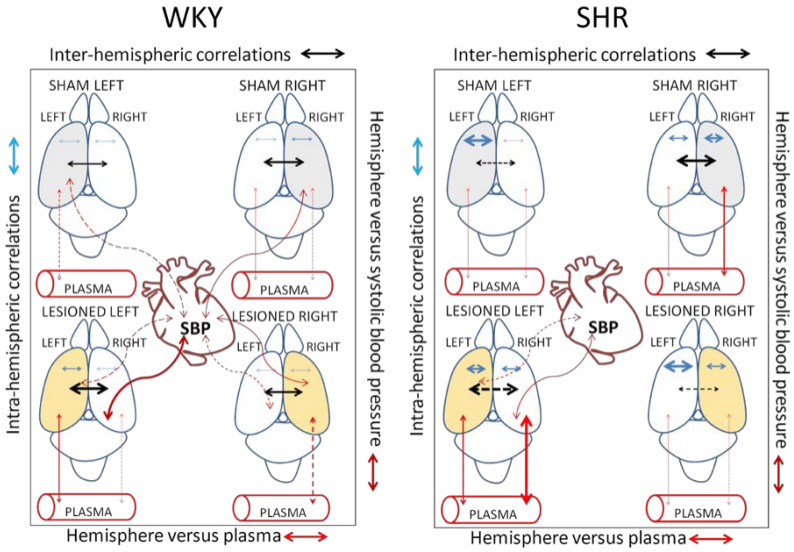
Schematic comparison of the main correlations in WKY and SHR. Intra-hemispheric correlations: blue arrows. Inter-hemispheric correlations: black arrows. Left or right hemisphere versus systolic blood pressure correlations: dark orange. Left or right hemisphere versus plasma correlations: red arrows. Continuous lines: positive correlations. Dotted lines: negative correlations. The thickness of lines is proportional to the number of correlations and the degree of significance.

**Table 1 biomedicines-10-02457-t001:** Intra- and inter-hemispheric correlations between brain aminopeptidase activities; between plasma aminopeptidase activities; between plasma aminopeptidase activities versus brain aminopeptidase activities and between aminopeptidase activities versus systolic blood pressure (SBP) in sham left spontaneously hypertensive rats (SHR).

**Sham Left SHR**		
**Intra-Hemispheric**		**Inter-Hemispheric**
**Sham Left (Left vs. Left)**	**Sham Left (Right vs. Right)**	**Sham Left (Left vs. Right)**
LPAla vs. LPCys r = +0.869 *p* = 0.001 RPAla vs. RPAsp r = +0.815 *p* = 0.004 LPCys vs. LPGlu r = +0.658 *p* = 0.03 RPAsp vs. RPCys r = +0.789 *p* = 0.006 LPAsp vs. RPAsp r = +0.699 *p* = 0.02 RACys vs. RAGlu r = +0.872 *p* = 0.001 LAAla vs. LAGlu r = +0.922 *p* = 0.0001 *RHAsp vs. RHCys r = −0.838 p = 0.002* LAAsp vs. LAGlu r = +0.916 *p* = 0.0002 RPAsp vs. RHAla r = +0.637 *p* = 0.04 LHAla vs. LHAsp r = +0.806 *p* = 0.004 LHAla vs. LHCys r = +0.744 *p* = 0.01 LHAsp vs. LHGlu r = +0.722 *p* = 0.01	LPAla vs. LHCys r = +0.683 *p* = 0.02 LPAsp vs. LHAla r = +0.630 *p* = 0.05 LPAsp vs. LHCys r = +0.755 *p* = 0.01 *LPAsp vs. LHGlu r = −0.794 p = 0.006* LPCys vs. LHCys r = +0.662 *p* = 0.03 LPGlu vs. LHCys r = +0.660 *p* = 0.03	LPAsp vs. RPAla r = +0.614 *p* = 0.05 LPAsp vs. RPAsp r = +0.699 *p* = 0.02 *LACys vs. RACys r = −0.628 *p* = 0.05* *LPAla vs. RAGlu r = −0.649 p = 0.04* LHAla vs. RPAla r = +0.647 *p* = 0.04 *LHAla vs. RACys r = −0.722 p = 0.01* *LHAla vs. RACys r = −0.722 p = 0.01* *LHAla vs. RAGlu r = −0.754 p = 0.01* *LHAla vs. RAGlu r = −0.754 p = 0.01* *LHCys vs. RAGlu r = −0.647 p = 0.04*
**Left Side vs. Plasma**	**Right Side vs. Plasma**	**Plasma vs. Plasma**
*LACys vs. PlAla r = −0.615 p = 0.05* LHAsp vs. PlGlu r = +0.718 *p* = 0.01 LHCys vs. PlCys r = +0.678 *p* = 0.03	*RAAla vs. PlAsp r = −0.649 p = 0.04* *RAGlu vs. PlCys r = −0.721 p = 0.01* *RHGlu vs. PlGlu r = −0.703 p = 0.02*	No correlations
**Left Side vs. SBP**	**Right Side vs. SBP**	
No correlations	No correlations	

The *p*-values and Pearson’s correlation coefficients (r) are indicated. Negative correlations are in blue italics. Positive correlations are in red. (L) left, (R) right, (H) hippocampus, (P) medial prefrontal cortex, (A) amygdala, (Glu) GluAP, (Ala) AlaAP, (Cys) CysAP and (Asp) AspAP.

**Table 2 biomedicines-10-02457-t002:** Intra- and inter-hemispheric correlations between brain aminopeptidase activities; between plasma aminopeptidase activities; between plasma aminopeptidase activities versus brain aminopeptidase activities and between aminopeptidase activities versus systolic blood pressure (SBP) in sham right spontaneously hypertensive rats (SHR).

**Sham Right SHR**		
**Intra-Hemispheric**		**Inter-Hemispheric**
**Sham Right (Left vs. Left)**	**Sham Right (Right vs. Right)**	**Sham Right (Left vs. Right)**
LAAla vs. LAGlu r = +0.757 *p* = 0.01 LAAsp vs. LAGlu r = +0.826 *p* = 0.003 LACys vs. LAGlu r = +0.674 *p* = 0.03 LHAla vs. LHCys r = +0.702 *p* = 0.02 LHCys vs. LHGlu r = +0.658 *p* = 0.03 LAAsp vs. LHAsp r = +0.927 *p* = 0.0001 LAGlu vs. LHAsp r = +0.747 *p* = 0.01 *LPAsp vs. LHGlu r = −0.700 p = 0.02* *LPGlu vs. LACys r = −0.780 p = 0.007*	RPAla vs. RPAsp r = +0.620 *p* = 0.05 RPAla vs. RPCys r = +0.889 *p* = 0.0006 RAAla vs. RAAsp r = +0.936 *p* = <0.0001 RAAla vs. RACys r = +0.920 *p* = 0.0002 RAAla vs. RAGlu r = +0.870 *p* = 0.001 RAAsp vs. RACys r = +0.908 *p* = 0.0003 RAAsp vs. RAGlu r = +0.842 *p* = 0.002 RACys vs. RAGlu r = +0.931 *p* = <0.0001 RPAsp vs. RHCys r = +0.836 *p* = 0.002 RPGlu vs. RAAla r = +0.718 *p* = 0.01 RPGlu vs. RAAsp r = +0.736 *p* = 0.01 RPGlu vs. RACys r = +0.681 *p* = 0.02	LAAla vs. RAAla r = +0.638 *p* = 0.04 LAAsp vs. RAAla r = +0.917 *p* = 0.0002 LAAsp vs. RAAsp r = +0.852 *p* = 0.001 LAAsp vs. RACys r = +0.852 *p* = 0.001 LAAsp vs. RAGlu r = +0.902 *p* = 0.0004 LAGlu vs. RAAla r = +0.888 *p* = 0.0006 LAGlu vs. RAAsp r = +0.830 *p* = 0.002 LAGlu vs. RACys r = +0.762 *p* = 0.01 LAGlu vs. RAGlu r = +0.780 *p* = 0.007 LHAsp vs. RAAla r = +0.892 *p* = 0.005 LHAsp vs. RAAsp r = +0.850 *p* = 0.001 LHAsp vs. RACys r = +0.940 *p* = <0.0001 LHAsp vs. RAGlu r = +0.907 *p* = 0.0003 LHCys vs. RPAla r = +0.648 *p* = 0.04 LHGlu vs. RPAla r = +0.674 *p* = 0.03 LPGlu vs. RHGlu r = +0.679 *p* = 0.03 LAAla vs. RPGlu r = +0.775 *p* = 0.008 *LPAla vs. RHAla r = −0.784 p = 0.007* *LPAsp vs. RHAla r = −0.719 p = 0.01*
**Left Side vs. Plasma**	**Right Side vs. Plasma**	**Plasma vs. Plasma**
LPAla vs. PlAla r = +0.706 *p* = 0.02 LACys vs. PlGlu r = +0.703 *p* = 0.02	RPGlu vs. PlAla r = +0.763 *p* = 0.01 RAAla vs. PlAla r = +0.660 *p* = 0.03 RAAsp vs. PlAla r = +0.811 *p* = 0.004 RACys vs. PlAla r = +0.678 *p* = 0.03 RPAsp vs. PlAsp r = +0.698 *p* = 0.02 RAAsp vs. PlCys r = +0.740 *p* = 0.01 RACys vs. PlCys r = +0.693 *p* = 0.02 RAGlu vs. PlCys r = +0.700 *p* = 0.02	PlAla vs. PlCys r = +0.842 *p* = 0.002
**Left Side vs. SBP**	**Right Side vs. SBP**	
No correlations	No correlations	

The *p*-values and Pearson’s correlation coefficients (r) are indicated. Negative correlations are in blue italics. Positive correlations are in red. (L) left, (R) right, (H) hippocampus, (P) medial prefrontal cortex, (A) amygdala, (Glu) GluAP, (Ala) AlaAP, (Cys) CysAP and (Asp) AspAP.

**Table 3 biomedicines-10-02457-t003:** Intra- and inter-hemispheric correlations between brain aminopeptidase activities; between plasma aminopeptidase activities; between plasma aminopeptidase activities versus brain aminopeptidase activities and between aminopeptidase activities versus systolic blood pressure (SBP) in lesioned left spontaneously hypertensive rats (SHR).

**Lesioned left SHR**		
**Intra-Hemispheric**		**Inter-Hemispheric**
**Lesioned Left (Left vs. Left)**	**Lesioned Left (Right vs. Right)**	**Lesioned Left (Left vs. Right)**
LPAla vs. LPCys r = +0.674 *p* = 0.03 LPCys vs. LPGlu r = +0.689 *p* = 0.02 LHAla vs. LHAsp r = +0.923 *p* = 0.0001 LHAla vs. LHCys r = +0.975 *p* = <0.0001 LHAla vs. LHGlu r = +0.930 *p* = 0.0001 LHAsp vs. LHCys r = +0.856 *p* = 0.001 LHAsp vs. LHGlu r = +0.946 *p* = <0.0001 LHCys vs. LHGlu r = +0.901 *p* = 0.0004 *LACys vs. LHAla r = −0.673 p = 0.03* *LACys vs. LHAsp r = −0.646 p = 0.04* *LACys vs. LHGlu r = −0.749 p = 0.01*	LPAla vs. LPCys r = +0.674 *p* = 0.03 RPAsp vs. RPCys r = +0.639 *p* = 0.04 RPAsp vs. RPGlu r = +0.616 *p* = 0.05 RPCys vs. RPGlu r = +0.808 *p* = 0.004 RHAla vs. RHCys r = +0.906 *p* = 0.0003 RPAla vs. RHAla r = +0.650 *p* = 0.04 RPAsp vs. RACys r = +0.782 *p* = 0.007 *RAGlu vs. RHGlu r = −0.739 p = 0.01*	LPAsp vs. RPGlu r = +0.672 *p* = 0.03 *LPGlu vs. RPAsp r = −0.629 p = 0.05* *LPGlu vs. RPCys r = −0.636 p = 0.04* LAAla vs. RAAla r = +0.662 *p* = 0.03 *LAAsp vs. RAGlu r = −0.703 p = 0.02* LAGlu vs. RAAsp r = +0.620 *p* = 0.05 *LAGlu vs. RAGlu r = −0.658 p = 0.03* *LPAsp vs. RAAsp r = −0.627 p = 0.05* *LPCys vs. RACys r = −0.709 p = 0.02* *LPGlu vs. RAAla r = −0.679 p = 0.03* *LPGlu vsRACys r = −0.803 p = 0.005* LAAla vs. RPGlu r = +0.631 *p* = 0.05 LACys vs. RHAla r = +0.654 *p* = 0.04 LACys vs. RHCys r = +0.625 *p* = 0.05
**Left Side vs. Plasma**	**Right Side vs. Plasma**	**Plasma vs. Plasma**
LPAsp vs. PlAla r = +0.751 *p* = 0.01 LPAsp vs. PlAsp r = +0.660 *p* = 0.03 LPAsp vs. PlCys r = +0.700 *p* = 0.0 LPAsp vs. PlGlu r = +0.618 *p* = 0.05 LAAla vs. PlAsp r = +0.670 *p* = 0.03 LAAla vs. PlCys r = +0.670 *p* = 0.03 LAAla vs. PlGlu r = +0.672 *p* = 0.03	RPAla vs. PlAla r = +0.617 *p* = 0.05 RPCys vs. PlAla r = +0.849 *p* = 0.001 RPCys vs. PlAsp r = +0.898 *p* = 0.0004 RPCys vs. PlCys r = +0.741 *p* = 0.01 RPCys vs. PlGlu r = +0.907 *p* = 0.0003 RPGlu vs. PlAla r = +0.919 *p* = 0.0002 RPGlu vs. PlAsp r = +0.933 *p* = <0.0001 RPGlu vs. PlCys r = +0.859 *p* = 0.001 RPGlu vs. PlGlu r = +0.936 *p* = <0.0001	PlAla vs. PlAsp r = +0.926 *p* = 0.0001 PlAla vs. PlCys r = +0.892 *p* = 0.0005 PlAla vs. PlGlu r = +0.927 *p* = 0.0001 PlAsp vs. PlCys r = +0.905 *p* = 0.0003 PlAsp vs. PlGlu r = +0.995 *p* = <0.0001 PlCys vs. PlGlu r = +0.906 *p* = 0.0003
**Left Side vs. SBP**	**Right Side vs. SBP**	
* LAAsp vs. SBP r = −0.685 p = 0.02 * *LACys vs. SBP r = −0.806 p = 0.004* *LAGlu vs. SBP r = −0.650 p = 0.04*	RAGlu vs. SBP r = +0.754 *p* = 0.01	

The *p*-values and Pearson’s correlation coefficients (r) are indicated. Negative correlations are in blue italics. Positive correlations are in red. (L) left, (R) right, (H) hippocampus, (P) medial prefrontal cortex, (A) amygdala, (Glu) GluAP, (Ala) AlaAP, (Cys) CysAP and (Asp) AspAP.

**Table 4 biomedicines-10-02457-t004:** Intra- and inter-hemispheric correlations between brain aminopeptidase activities; between plasma aminopeptidase activities; between plasma aminopeptidase activities versus brain aminopeptidase activities and between aminopeptidase activities versus systolic blood pressure (SBP) in lesioned right spontaneously hypertensive rats (SHR).

**Lesioned Right SHR**		
**Intra-Hemispheric**		**Inter-Hemispheric**
**Lesioned Right (Left vs. Left)**	**Lesioned Right (Right vs. Right)**	**Lesioned Right (Left vs. Right)**
LPAla vs. LPAsp r = +0.769 *p* = 0.009 LPAla vs. LPCys r = +0.624 *p* = 0.05 LPAsp vs. LPCys r = +0.773 *p* = 0.008 LAAla vs. LAAsp r = +0.759 *p* = 0.01 LHAla vs. LHAsp r = +0.970 *p* = <0.0001 LHAla vs. LHCys r = +0.977 *p* = <0.0001 LHAla vs. LHGlu r = +0.989 *p* = <0.0001 LHAsp vs. LHCys r = +0.997 *p* = <0.0001 LHAsp vs. LHGlu r = +0.992 *p* = <0.0001 LHCys vs. LHGlu r = +0.993 *p* = <0.0001 LAGlu vs. LHAla r = +0.619 *p* = 0.05 LAGlu vs. LHAsp r = +0.646 *p* = 0.04 LAGlu vs. LHCys r = +0.630 *p* = 0.05	RHAsp vs. RHGlu r = +0.744 *p* = 0.01 RHCys vs. RHGlu r = +0.679 *p* = 0.03 RPAla vs. RAAsp r = +0.724 *p* = 0.01 *RPCys vs. RACys r = −0.683 p = 0.02* *RPGlu vs. RHAsp r = −0.790 p = 0.006* RAAsp vs. RHCys r = +0.698 *p* = 0.02 RAGlu vs. RHAsp r = +0.653 *p* = 0.02	LPCys vs. RPGlu r = +0.627 *p* = 0.05 LPGlu vs. RAGlu r = +0.635 *p* = 0.04 *LHAla vs. RAGlu r = −0.624 p = 0.05* *LHAsp vs. RAAla r = −0.636 p = 0.04* *LHGlu vs. RAAla r = −0.626 p = 0.05* *LHGlu vs. RAGlu r = −0.615 p = 0.05*
**Left Side vs. Plasma**	**Right Side vs. Plasma**	**Plasma vs. Plasma**
LPGlu vs. PlGlu r = +0.677 *p* = 0.03	*RHAsp vs. PlAla r = −0.638 p = 0.04* *RHAsp vs. PlCys r = −0.677 p = 0.03* *RHCys vs. PlCys r = −0.663 p = 0.03*	PlAla vs. PlCys r = +0.938 *p* = <0.0001 PlAla vs. PlGlu r = +0.734 *p* = 0.01 PlAsp vs. PlCys r = +0.744 *p* = 0.01 PlAsp vs. PlGlu r = +0.743 *p* = 0.01 PlCys vs. PlGlu r = +0.780 *p* = 0.007
Left Side vs. SBP	Right Side vs. SBP	
No correlations	No correlations	

The *p*-values and Pearson’s correlation coefficients (r) are indicated. Negative correlations are in blue italics. Positive correlations are in red. (L) left, (R) right, (H) hippocampus, (P) medial prefrontal cortex, (A) amygdala, (Glu) GluAP, (Ala) AlaAP, (Cys) CysAP and (Asp) AspAP.

## Data Availability

The data presented in this study are available on request from the corresponding author.

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
