# Peer review of "Aminopeptidase Activities Interact Asymmetrically between Brain, Plasma and Systolic Blood Pressure in Hypertensive Rats Unilaterally Depleted of Dopamine"

_biomedicines, 2022, doi:10.3390/biomedicines10102457_

Round 1

Reviewer 1 Report

This manuscript by Banegas et al is well-written, well-structured; the results and discussions are well presented and explained. I therefore recommend it for publication in Biomedicines.

Author Response

Dear Reviewer,

Thank you very much for your kind comments.

Best regards

Manuel Ramírez-Sánchez

Reviewer 2 Report

The author Banegas et al., add a step forward in the cerebral asymmetry of aminopeptidase activities and the model of dynamic asymmetrical neurovisceral integration by a well-experimental executed plan. This study elucidates an important issue about a selective unilateral depletion of a specific neurotransmitter, changes the bilateral pattern of brain area response, with modulation of bilateral response.

They showed results are relevant, and the step-by-step graphical representation of significant correlations between brain aminopeptidase activities, plasma aminopeptidase activities, and systolic blood pressure in the four groups of spontaneously hypertensive rats, aid the readers to connect these complex experimental paradigms.

This is a well-written research paper and the length is commensurate with the message. 

The literature cited is of important relevance. For the mentioned reasons, the article may be accepted for publication without revision.

Author Response

(The authors gave the same response as above.)

Reviewer 3 Report

Review on the manuscript titled ‘Aminopeptidase activities interact asymmetrically between brain cortico-limbic areas, plasma and systolic blood pressure in hypertensive rats unilaterally depleted of dopamine. A comparative study with normotensive rats’ by Banegas I et al., submitted to Biomedicines

Manuscript ID: biomedicines-1901772 

Dear Authors, 

In the present review entitled ‘Aminopeptidase activities interact asymmetrically between brain cortico-limbic areas, plasma and systolic blood pressure in hypertensive rats unilaterally depleted of dopamine. A comparative study with normotensive rats’, Banegas and colleagues investigated the interaction of the brain, plasma, and systolic blood pressure in spontaneously hypertensive rats.

The main strength of this manuscript is that it addresses an interesting and timely topic, suggesting an important role for the asymmetrical function of the amygdala in cardiovascular control.

In general, I think the idea of this study is really interesting and the authors’ fascinating observation and interpretation on this timely topic may be of interest to the readers of Biomedicines. However, some comments, as well as some crucial evidence that should be included to support the authors’ argumentation, needed to be addressed to improve the quality of the manuscript, its adequacy, and its readability prior to the publication in the present form. My overall opinion is to publish this research article after the authors have carefully considered my suggestions below, particularly reshaping parts of the Introduction section by adding more evidence.

Comments:

1.     Title: The title reaches four lines. I suggest shortening the title presenting the most important findings and the conclusion of this manuscript.

2.     Abstract: Please clearly and proportionally present the background, the methods, the results, and the conclusion proportionally with 200 words. Also, the elements of ‘aminopeptidase’, ‘asymmetry’, ‘emotional processing’, ‘hypertension’, and ‘dopamine’ are sparsely presented, and the relationship is not well presented. I recommend that the authors concisely present the connection, leading to the rationale of this study in the background.

3.     Keyword: Please list ten keywords and use as many keywords as possible in the first two sentences of the abstract.

4.     A Graphical Abstract is highly recommended.

5.     In general, I recommend that the authors use more evidence to back their claims, especially in the Introduction of the article, which I believe is currently lacking. Thus, I suggest deepening the subject of the manuscript, as the bibliography is too concise. In my opinion, less than 60-70 references for an original article are insufficient. Therefore, I suggest focusing their efforts on researching and presenting more relevant literature: I believe that adding more studies and reviews will help provide better and more accurate background to this manuscript.

6.     Introduction: The authors attempted to introduce the main elements, ‘aminopeptidase’, ‘asymmetry’, ‘emotional processing’, ‘hypertension’, and ‘dopamine’ in this section; however, the descriptions of each element are not presented in a logical sequence, in a way to connect them, and thus not leading well to the rationale and the objectives of this study. Furthermore, the authors did not conduct behavioral tests for emotional domain, while mentioning emotional domain. I recommend that the authors reorganize this section, focusing on the most important elements and their connection. In addition, subcortical dopamine has a strong influence on other behavioral domains such as cognition, possessing a neural correlate to the prefrontal cortex. Spontaneously hypertensive rat is a frequently used animal model of schizophrenia (https://doi.org/10.3390/jcm10091809; https://doi.org/10.1038/s41380-021-01326-4; https://doi.org/10.3390/biomedicines9030235).

7.     Figure 1: I am not certain if the presentation of the figure is relevant to the introduction.

8.     Discussion: Based on the results, I recommend that the authors present this section to fully expand arguments including the weakness, the limitations, the potential of this review, the goal, the challenge, the knowledge, and the technology necessary to achieve this goal, and future research direction, among others.

9.     Conclusion: In this section I recommend that the authors present the take home message as the core part of this manuscript. Thus, please provide a synthesis of the data presented in the paper as well as possible keys to advance research.

Overall, the manuscript contains no figure, one table and 54 references. I believe that this manuscript might carry important value studying the interaction of the brain, plasma, and systolic blood pressure in spontaneously hypertensive rats. I hope that, after these careful revisions, the manuscript can meet the Journal’s high standards for publication.

Best regards,

Reviewer

Author Response

Dear Reviewer,

-Thank you very much for your kind comments. Your suggestions are very appropriate and I have tried to attend to all of them, mainly in the title, abstract, keywords, introduction and bibliography. Giving a coherent structure to the different aspects that underlie the study (dopamine, limbic system structures, asymmetry, hypertension, renin-angiotensin system, aminopeptidases) and combining it with the lace in the special number of "10th Anniversary of Biomedicines & mdash; Advances in Proteinases and Proteinase Inhibitors" is complicated.

-I have also performed a Graphical Abstract with the most interesting results.

-I have modified the summary giving it the logical order that you suggest, based on the new keywords that I have included. Based on the above, I have added a first paragraph to the introduction giving it that logical coherence and including new relevant bibliography. With that new introductory paragraph, the rest of the introduction expands on the previous points and positions the article within the context of the special issue on "10th Anniversary of Biomedicines & mdash; Advances in Proteinases and Proteinase Inhibitors".

-Regarding Figure 1, in my opinion it is useful for the reader not familiar with the subject. It is designed so that this reader can quickly visually grasp the importance of aminopeptidases in the renin-angiotensin system.

-Regarding the discussion and conclusions, I have highlighted the perhaps most interesting results, I have commented about some limitations and some possible keys to advance in the investigation as you suggested.

I expect you now find the manuscript appropriate for publication.

Many thanks

Manuel Ramírez-Sánchez

Round 2

Reviewer 3 Report

The 2nd Review on the manuscript titled ‘Aminopeptidase activities interact asymmetrically between brain cortico-limbic areas, plasma and systolic blood pressure in hypertensive rats unilaterally depleted of dopamine. A comparative study with normotensive rats’ by Banegas I et al., submitted to Biomedicines

Manuscript ID: biomedicines-1901772 

Dear Authors, 

I am pleased to see that the authors did an excellent work clarifying most of the comments I have raised in the previous round of the review session. Currently, this paper is a well-written and valuable piece of research presenting the interaction of the brain, plasma, and systolic blood pressure in spontaneously hypertensive rats. That said, I just suggest some minor points below, I believe, for the betterment of this manuscript to finalize my review session.

Comments:

1.     Introduction: All essential elements of this manuscript are now nicely introduced and presented in a logical sequence. Probably, the following literature may reinforce the richness of the context to provide an updated broader view in depth, which current readers can find relevance to their work in this manuscript, regarding, for example, dopamine in cognitive function, mitochondria dysfunction in schizophrenia and hypertension, and schizophrenia in dissociative model: https://doi.org/10.3390/ijms23073452; https://doi.org/10.3390/cells11162607; doi: 10.3389/fpsyt.2022.845493).

2.     Figures: I suggest presenting all figures in color.

Overall, the manuscript contains five figures, two tables, and 59 references. This is a timely and needed work, thus I believe that manuscript now meets the Journal’s standards for publication. 

Best regards,

Reviewer 

Author Response

Dear Reviewer:

Thank you very much for your comments and suggestions.

  1. The three articles that you suggest have been now included in the revised manuscript.
  2. All the figures are now in color.

Best regards,

Manuel Ramírez-Sánchez